# Impact of SARS-CoV-2 Alpha and Gamma Variants among Symptomatic Pregnant Women: A Two-Center Retrospective Cohort Study between France and Brazil

**DOI:** 10.3390/jcm11092663

**Published:** 2022-05-09

**Authors:** Elie Mosnino, Lisandra S. Bernardes, Jeremie Mattern, Bruna Hipólito Micheletti, Amarilis Aparecida de Castro Maldonado, Christelle Vauloup-Fellous, Florence Doucet-Populaire, Daniele De Luca, Alexandra Benachi, Alexandre J. Vivanti

**Affiliations:** 1Division of Obstetrics and Gynecology, Antoine Béclère Hospital, Paris Saclay University, Assistance-Publique des Hôpitaux de Paris, 92140 Clamart, France; elie.mosnino@aphp.fr (E.M.); jeremie.mattern@aphp.fr (J.M.); alexandra.benachi@aphp.fr (A.B.); 2Division of Obstetrics and Gynecology, Hospital e Maternidade Sepaco, São Paulo 04005-002, Brazil; lisbernardes@yahoo.com (L.S.B.); bhmicheletti@gmail.com (B.H.M.); amarilisacm@gmail.com (A.A.d.C.M.); 3School of Medicine, Faculdade Israelita de Ciências da Saude Albert Einstein, São Paulo 05521-200, Brazil; 4Faculdade de Medicina FMUSP, Universidade de Sao Paulo, São Paulo 05508-070, Brazil; 5Division of Virology, Paul Brousse Hospital, Paris Saclay University, Assistance-Publique des Hôpitaux de Paris, 94800 Villejuif, France; christelle.vauloup-fellous@aphp.fr; 6Research Group on Infections in Pregnancy (GRIG), 75000 Paris, France; 7Inserm U-1193, Paris Saclay University, Villejuif, 94800 Paris, France; 8Division of Microbiology, Antoine Béclère Hospital, Paris Saclay University, Assistance-Publique des Hôpitaux de Paris, 92140 Clamart, France; florence.doucet-populaire@aphp.fr; 9Division of Pediatrics and Neonatal Critical Care, Antoine Béclère Hospital, Paris Saclay University, Assistance-Publique des Hôpitaux de Paris, 92140 Clamart, France; daniele.deluca@aphp.fr

**Keywords:** SARS-CoV-2, COVID-19, variant, strain, pregnancy outcomes, pregnancy complications

## Abstract

New variants of SARS-CoV-2 are a major source of concern, especially for pregnant women and in the perinatal context. The primary aim of this study was to compare the severity of COVID-19 infection in pregnant women depending on strain predominance between wild-type Alpha and Gamma variants. The secondary aim was to study the impact of these strains on obstetrical and neonatal outcomes. We conducted a two-center international retrospective cohort study, which included two type III maternity hospitals, one in France and one in Brazil, comparing the first period corresponding to the wild-type strain and the second period corresponding to the predominance of the Alpha variant in France and the Gamma variant in Brazil. We included 151 pregnant women with symptomatic SARS-CoV-2 infection confirmed by RT-PCR. The rate of severe-to-critical infection, according to the WHO definition, was seven-fold higher in patients infected during the “variant period” than in patients infected during the “wild-type period” (aOR = 7.07, 95CI [2.50–21.6], *p* < 0.001). There were no statistical differences concerning composite obstetrical and neonatal outcomes between the different periods. While analyzing each variant separately, it was found that, in France, the risk of developing a severe-to-critical infection was three times greater during the Alpha period than during the wild-type period (OR = 3.25, 95CI [0.70–15.6], *p* = 0.13) and, in Brazil, the risk was twelve times greater during the Gamma period than during the wild-type period (OR = 11.8, 95CI [2.46–72.3], *p* = 0.003). The Alpha and Gamma variants of SARS-CoV-2 seem to be more dangerous in the obstetrical context. With the rapid emergence of new variants, it is necessary to accelerate vaccination to protect women and newborn children.

## 1. Introduction

Since the beginning of the COVID-19 pandemic caused by SARS-CoV-2 in 2020, particular attention has been paid to the obstetrical and perinatal contexts. The numerous data that have been collected in the meantime allow us to affirm that infections developed during pregnancy are more severe compared to those developed among non-pregnant women and that they require more oxygen therapy, admission to intensive care units, and mechanical ventilation [1,2,3]. Infected patients also seem to have more obstetrical complications: the latest data show an increased risk of prematurity (spontaneous or induced) and pre-eclampsia [2,3,4]. The risk of in utero fetal death, however, is still debated due to contradictory results from large national cohorts [5,6,7,8].

From the end of 2019, the COVID-19 epidemic, first occurring in China, spread rapidly around the world, leading to the first global wave in early 2020, which caused millions of infections and deaths. While the pandemic seemed to be subsiding, successive epidemic resurgences took place, notably with the appearance of new variants, some of which are defined as variants of concern (VOCs) by the World Health Organization (WHO) [9] since they are associated with increased transmissibility; increased virulence; and a reduced immune response, particularly to vaccines [10,11]. At the end of 2020, the B.1.1.7 variant (WHO label “Alpha”) first emerged in the United Kingdom and the B.1.351 variant (WHO label “Beta”) in South Africa. Then, in Brazil, the variant P.1 (WHO label “Gamma”) appeared, and in India, the variant B.1.617.2 (WHO label “Delta”) appeared before the appearance of the variant B.1.1.529 in November 2021, identified for the first time in South Africa (WHO label “Omicron”).

Some authors suggest differences in pathogenicity between the different epidemic waves linked to the different variants, affecting younger adults [12] or leading to more severe forms [7,13,14]. Ong et al. in Singapore found more severe forms linked to the Delta variant [15].

Few studies have compared the direct impact of different COVID-19 variants in the obstetrical context. Some registries and case series first suggested an increase in severe cases between the different waves, possibly linked to the emergence of VOCs [12,16,17,18,19]: this is, for example, the case in the United Kingdom (UK), where a referral center for extracorporeal membrane oxygenation (ECMO) was, in proportion to the number of total requests, subject to three times more ECMO requests in pregnant women during the second wave, linked to the Alpha variant, than during the first wave [17]. In the United States, the appearance of the Delta variant led to a proportionally greater increase in the number of severe cases in pregnant women [18]. The problem with these descriptive registries is that they do not distinguish between a change in pathogenicity and a difference in the distribution of infections, with possibly more pregnant women being infected in successive waves of VOCs.

Few studies have assessed the impact of variants on pregnancy. In India, a single-center retrospective cohort study showed an increase in the severity of infections related to the second wave with the emergence of the Delta variant (5.7% of deaths versus 0.7% and 11.6% of ICU admissions versus 2.4% in a total of 1500 included patients) [20]. In Europe, two large national registries confirm the increased virulence of the Alpha and Gamma variants during pregnancy [5,7]. Only one of these four studies examined the impact of the Gamma variant. In Brazil, using a national SARS-CoV-2 register, Gonçalves et al. showed that the maternal death rate doubled in 2021, when the Gamma variant predominated, compared to 2020, when the wild-type predominated, and that the only significant change between these two periods was the appearance of the new variant [21].

The aim of our study was to determine whether there are differences in pathogenicity between the Alpha, Gamma, and wild-type variants of SARS-CoV-2 in the obstetrical setting.

## 2. Materials and Methods

### 2.1. Study Design

We conducted a two-center, international, retrospective, cohort study, which included two type III maternity hospitals, one in France (Antoine Béclère Hospital in Clamart) and one in Brazil (Sepaco Hospital in São Paulo), comparing the first period from the beginning of the COVID-19 epidemic in February 2020 until 13 February 2021, corresponding to the wild-type strain, and the second period from 14 February 2021 to 1 June 2021, corresponding to the period when the Alpha variant in France and the Gamma variant in Brazil became dominant. The study population included all pregnant women who presented a symptomatic infection with SARS-CoV-2 during pregnancy, confirmed by RT-PCR on a nasopharyngeal swab, and who delivered before 1 June 2021. We did not include patients who had suffered a miscarriage in the first trimester or patients who had not given birth at the end of the collection period. Patients who had undergone a medical termination of pregnancy, patients who were postpartum at the time of diagnosis of infection, patients who were lost to follow-up after diagnosis of infection, and asymptomatic patients were excluded. Indeed, as the methods of carrying out RT-PCR (screening and symptoms) differed between the two centers and according to the different epidemic waves, we chose not to include asymptomatic patients tested on a systematic basis during scheduled hospitalizations or at the time of delivery.

### 2.2. Epidemiology of Variants in France and Brazil during the Study Period

The speed of establishment of each variant and its rapid predominance combined with the fact that not all samples were sequenced pushed us not to study the variants directly but the corresponding periods.

During the analysis period, in France, Santé Publique France (SPF) took responsibility for collecting data on the variants by sequencing the genome of biweekly-sampled positive PCRs from 7 January 2021 through Flash surveys [22]. The first case of a SARS-CoV-2 variant was detected on 26 December 2020 in France. Until February 2021, the wild-type strain was in the majority. As of 14 February 2021, the Alpha variant became the majority in France, representing approximately 80% of the infections in the period up to 1 June 2021 [22].

In Brazil, the first case of a SARS-CoV-2 variant was detected in November 2020. Until February 2021, the wild-type strain was in the majority. From February 2021, the Gamma variant in Brazil accounted for most infections in the period up to 1 June 2021. The available data are based on the paper by Martins et al., which showed that, by the end of February 2021, the overwhelming majority of tests sequenced indicated the Gamma variant [23], and they are corroborated by the open-source SARS-CoV-2 strain data from the collaborative group GISAID [24].

We therefore chose to compare our study population from 1 February 2020 to 13 February 2021 when the majority strain in Brazil and France was the wild type, and then from 14 February 2021 to 1 June 2021 when the majority strains were the Alpha variant in France and the Gamma variant in Brazil.

### 2.3. Data

Data were retrospectively extracted from patients’ medical records for the present study after anonymization and collected at the study coordination center. Data collected included maternal age, geographical origin, body mass index (BMI), parity, smoking status, pre-existing chronic hypertension, pre-existing diabetes mellitus, pre-existing pulmonary, renal or liver disease, date of RT-PCR, hospital admission in relation to SARS-CoV-2, and all data necessary for the outcomes defined below.

### 2.4. Outcomes

The primary outcome was a composite endpoint of severe-to-critical infection according to the WHO classification [25]. It included the following: need for oxygen therapy, admission to the intensive care unit (ICU), acute respiratory distress syndrome (ARDS), mechanical ventilation, ECMO, and maternal death. The presence of pneumonia identified by chest computed tomography (CT) was also collected as a secondary outcome, but it was not included in the composite outcome.

As secondary outcomes and exploratory measures, obstetric and neonatal outcomes were analyzed by composite criteria of obstetric and neonatal events, previously defined by Badr et al. [4]. The composite adverse obstetric outcome was defined as preterm delivery (<37 WG); pre-eclampsia; eclampsia or hemolysis, elevated liver enzymes, and low platelet count (HELLP) syndrome; unscheduled cesarean delivery; deep venous thrombosis; pulmonary embolism; pregnancy loss (<24 WG); intrauterine fetal demise (>24 WG); or maternal death. The composite adverse neonatal outcome was defined as small for gestational age (SGA; birthweight Z-score < −1.28), neonatal intensive care unit (NICU) admission, 5′ Apgar score <7, respiratory distress, grade 3/4 intraventricular hemorrhage, or neonatal death.

Obstetrical and neonatal secondary outcomes of the study included each outcome of the composite variables, as well as delivery at <32 weeks, spontaneous delivery at <37 weeks, suspected fetal distress (category II or III fetal heart rate tracing [26]), cesarean delivery, postpartum hemorrhage (defined as blood loss of >500 mL), umbilical artery pH, and congenital abnormalities.

Finally, we examined the impact of each variant within each country in a secondary subgroup analysis according to primary and secondary outcomes.

### 2.5. Statistical Analysis

The baseline characteristics of the study participants were described and analyzed by comparing the proportions in each group using a Chi2 test for categorical variables, replaced, if necessary, by Fisher’s exact test. Quantitative variables were analyzed by comparing ranks using the Mann–Whitney U test. The results are given as medians and IQR.

Maternal outcomes were compared between groups using multivariable logistic regression analysis adjusted for age, obesity (BMI > 30), center, pre-existing condition status (yes or no), and geographic origin. Neonatal outcomes were adjusted for geographic origin, preterm birth, and center. Quantitative outcomes were obtained after linear regression with adjustment for the same variables.

There were no statistical interactions between variant types and adjusting variables (including center). The threshold of statistical significance was set at *p* < 0.05. Mixed models were not better in sensitivity analyses.

All analyses were performed with R version 4.01 (RCore Team 2020, Vienna, Austria) using the package GTSummary [27].

### 2.6. Ethical Approval

The study was approved by the appropriate ethical board for each recruiting center (CEROG 2021-OBST-0503 and IRB 36855220.6.0000.0086), and informed consent was obtained when required by the relevant local regulations.

## 3. Results

### 3.1. Baseline Characteristics

In total, among the 238 SARS-CoV-2 PCR-positive pregnant women, we identified 151 patients who had developed symptomatic COVID-19 infection since the beginning of the epidemic and who delivered before 1 June 2021 (Figure 1). Of these, 126 were diagnosed before 13 February 2021, thus constituting the patients considered to be infected with the wild-type strain of COVID-19 in both countries (68 patients in France, and 58 patients in Brazil), and 25 patients between 14 February and 1 June 2021, corresponding to the period when the variants became predominant, that is, Alpha in France (11 patients included) and Gamma in Brazil (14 patients included).

The baseline characteristics are presented in Table 1. For each period, approximately the same number of patients was included in each of the two study centers. The mean age of the patients was 31 years (IQR: 28–35). Among all baseline characteristics, the only variable that differed significantly was pre-pregnancy BMI, which was 26 kg/m^2^ (IQR: 23–29.9) in the “wild-type period” versus 29 (26–35) in the “variant period”. There was no difference in the distribution of the other variables, particularly among the possible confounders of ethnicity, age, and medical history (Table 1). The basic characteristics of the patients detailed by country are presented in the Appendix A.

### 3.2. Primary Outcome—Disease Severity

The rate of severe-to-critical infection according to the WHO definition was significantly higher in patients infected during the “variant period” than in patients infected during the “wild-type period” (aOR = 7.07, 95CI [2.50–21.6], *p* < 0.001) (Table 2). All secondary endpoints significantly increased during the “variant period” compared to the “wild-type period”, especially the rate of ICU admission, which increased from 10% to 32% (aOR = 3.75, 95CI [1.22–11.2], *p* = 0.007), and the rate of mechanical ventilation, which increased from 4.8% to 20% (*p* = 0.014). No patient received ECMO. One maternal death occurred due to COVID-19 in Brazil during the COVID-19 variant period.

### 3.3. Secondary Outcomes—Obstetrical and Neonatal Outcomes

There was no significant difference in the composite obstetrical event criterion (aOR = 1.86, 95CI [0.73–4.86], *p* = 0.2) (Table 3). There was no significant difference in the distribution of most secondary obstetric outcomes between patients infected during the “wild-type period” and patients infected during the “variant period”. The only significant difference was in the number of preterm births before 37 weeks, which increased in the “variant period” (aOR = 3.87, 95CI [1.44–10.6], *p* = 0.007). The rate of unscheduled cesarean sections was not significantly increased between the two periods (40% versus 33%, *p* = 0.8). Three in utero fetal demises or late miscarriages occurred in the “wild-type period” (3/126 = 2.4%), one attributed to inaugural placental abruptio, one attributed to cervical incompetence, and one attributable to COVID-19-related placental damage, and one in utero fetal demise occurred in the “variant period” (1/25 = 4.0%), which was attributable to COVID-19-related placental damage (Table 3).

A total of 155 neonates were included in the analysis of neonatal outcomes. We found no significant difference in the composite neonatal event criterion between the two periods (aOR = 1.42, 95CI [0.55–3.58], *p* = 0.5) (Table 4). There was no significant difference in the distribution of most secondary neonatal outcomes between neonates of mothers infected during the “wild-type period” and neonates of mothers infected during the “variant period”. The only significant difference was in the number of NICU admissions (aOR = 4.94, 95CI [1.37–18.4], *p* = 0.014). This could be related to the significantly increased number of preterm births before 37 WG and to the rate of neonatal respiratory distress, which was higher in the “variant period” than in the “wild-type period”, but not significantly (26% versus 7.8% aOR = 3.34 95CI [0.88–12.2] *p* = 0.07). There were four neonatal deaths during the “wild-type period” (4/128 = 3.1%) and none during the “variant period”. The four deaths were due to the following reasons: one due to a premature birth induced at 35 weeks in a child with polymalformative syndrome and severe growth restriction, one due to a severe congenital heart defect that could not be managed at birth, one due to chorioamnionitis at 31 weeks, and one due to extreme prematurity (26 weeks) in a patient who required an emergency cesarean delivery for maternal rescue because of a severe COVID-19 infection (Table 4).

### 3.4. Subpopulation Analysis: Comparison of Each Variant in Each Center

In a secondary analysis, we looked at the impact of each variant within each center. In Clamart (France), there were 79 patients with COVID-19 recruited in our study, 68 in the “wild-type period” and 11 in the “variant period” corresponding to the predominance of the Alpha variant in France (Figure 1 flowchart and Appendix A). The risk of developing a severe-to-critical infection during pregnancy during the Alpha variant period compared to the wild-type period was approximately three times greater (OR = 3.25, 95CI [0.70–15.6], *p* = 0.13) (Appendix A) There was no significant difference between the two periods for composite obstetrical (OR = 2.45, 95CI [0.63–10.8], *p* = 0.2) and neonatal outcomes (OR = 0.5, 95CI [0.09–2.17], *p* = 0.4) (Appendix A).

In São Paulo (Brazil), there were 72 patients with COVID-19 recruited during the study period, 58 in the “wild-type period” and 14 in the “variant period” corresponding to the predominance of the Gamma variant in Brazil (Figure 1 flowchart and Appendix A). The risk of developing a severe-to-critical infection during pregnancy during the Gamma variant period compared to the wild-type period was approximately twelve times greater (OR = 11.8, 95CI [2.46–72.3], *p* = 0.003) (Appendix A). There was no significant difference between the two periods for composite obstetrical (OR = 1.06, 95CI [0.23–4.87], *p* > 0.9) and neonatal (OR = 2.69, 95CI [0.77–9.42], *p* = 0.12) outcomes (Appendix A).

## 4. Discussion

This study indicates an increase in pathogenicity of the Alpha and Gamma variants compared to wild-type COVID-19 in the obstetrical setting. In symptomatic COVID-19-infected pregnant women, the risk of developing a severe-to-critical form of COVID-19 during the “variant period” (Alpha and Gamma) compared to the “wild-type period” increased seven-fold (aOR = 7.07, 95CI [2.50–21.6], *p* < 0.001). Maternity wards are subject to many questions regarding the impact of these variants, and few studies are available to date.

Concerning the Gamma variant, Gonçalves et al. found a two-fold increase in maternal deaths related to SARS-CoV-2 in 2021 compared to 2020 (17.4% vs. 7.5%, OR = 2.6 95CI [2.28–2.97]) [21]. Moreover, ICU admission and intubation increased significantly in 2021 compared to 2020 (respectively OR = 1.6 95CI [1.45–1.77] and OR = 2.02 [1.78–2.30]) [21]. For the Alpha variant, it has been suggested that there is an increase in severe forms of COVID-19, with a multiplication of the risks of the severe forms by about 2 to 3. This is notably the case in Italy; Donati et al. used an Italian national registry (ItOSS) with data up to June 2021, which included more than 3000 hospitalized pregnant women, and concluded the following on the basis of their exhaustive Italian register: “The need for ventilatory support and/or ICU admission among women with pneumonia increased during the Alpha compared to the wild-type period (3.24, 1.99–5.28) [5].” Similarly, in the UK, Vousden et al., using the UKOSS national cohort analyzed up to July 2021, which also included more than 3000 pregnant women infected with SARS-CoV-2, concluded that “the proportion that experienced moderate to severe infection significantly increased between wild-type and Alpha periods (24.4% vs. 35.8%; aOR = 1.75, 95CI [1.48–2.06])” [7].

The current dynamics of variant evolution are extremely rapid. The second half of 2021 was marked by the appearance of the Delta variant. Vousden et al. showed that “the proportion that experienced moderate to severe infection significantly increased […] between Alpha and Delta periods (35.8% vs. 45.0%; aOR1.53, 95%CI 1.07–2.17)”. Seaseley et al. recently found similar results [19]. Concerning obstetrical and neonatal outcomes, it is unclear if the Delta variant increases poor outcomes. One recent national register in the USA found an increased risk of stillbirth during the Delta period (aRR = 4.04 95CI [3.28–4.97]; *p*-value for effect modification by period (pre-Delta period versus period of Delta predominance): <0.001) [8]. These results are not consistent with those from the Italian and UK registers, which found no differences concerning stillbirth or neonatal and maternal deaths.

Our study has several strengths. First, there are few studies that have looked at the impact of variants in the obstetrical context, especially for the Gamma variant for which there is only one maternal death register available. This is a multicenter and country study. Coincidentally, the time periods between wild-type and variant occurrence in the two countries were precisely superimposed. This reduced the time effect bias and made the different populations more comparable. The fact that only symptomatic patients were selected meant that screening policies, which differed from country to country and evolved with the different epidemic waves, could be avoided.

Our study also has several weaknesses. The impact of the different variants on obstetrical and neonatal outcomes is less clear in our study. There seems to be a tendency for obstetrical and neonatal outcomes to worsen, which is consistent with the study conducted by Badr et al. [4], but we might suffer from a lack of power due to the small number of patients included. Moreover, practices are different between France and Brazil. We therefore avoided taking into consideration criteria for which the underlying obstetrical practices are different between the countries (e.g., the rate of cesarean section, which is therefore not included in our judgment criteria).

We chose not to study the variants directly but the corresponding periods. This is consistent with the speed of establishment of each variant and its rapid predominance. In addition, not all patient samples were sequenced. This method is widely used by authors who study different strains of SARS-CoV-2 and more generally by those who study infectious diseases.

We did not include data on vaccination in our data collection, as it was only in the deployment phase in both countries at the time that the data collection was completed. In France, on 1 June, less than 9% of women under 50 years old were fully vaccinated [28] and the recommendations for vaccination in pregnant women had only just been issued. In Brazil, on 1 June 2021, 10.6% of the population was fully vaccinated [29], and this concerned mainly pensioners and health professionals because of the lack of accessibility to vaccination at that time.

## 5. Conclusions

The severity of COVID-19 infections in pregnant women seems to have been clearly aggravated by the appearance of the Alpha and Gamma variants. It is essential to remain vigilant regarding the consequences that future variants will have in the obstetrical context so as to adapt the management of pregnant women in public health policies. Further studies are needed to confirm the impact of the different variants on pregnancy. At the same time, vaccination seems to be the only convincing bulwark that we currently have at our disposal to limit the severe forms of the disease, particularly in the obstetrical context.

## Figures and Tables

**Figure 1 jcm-11-02663-f001:**
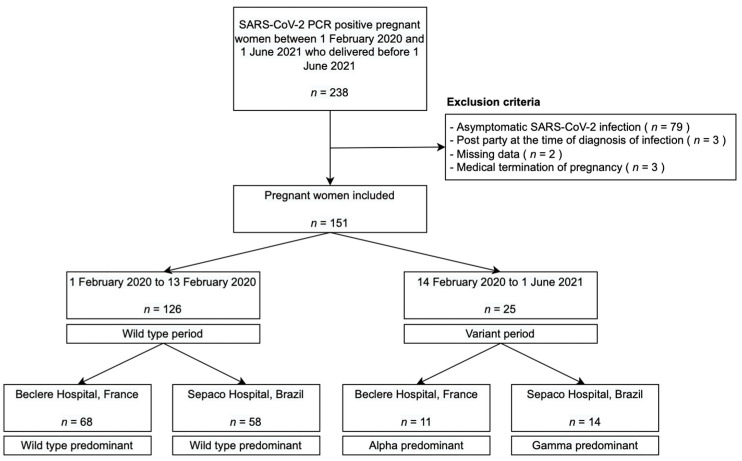
Flowchart.

**Table 1 jcm-11-02663-t001:** Baseline characteristics of COVID-19 symptomatic pregnant women during “wild-type period” and “variant period”.

Characteristic	Overall 1 February 2020–1 June 2021	Wild-Type Period 1 February 2020–13 February 2021	Variant Period 14 February 2021–1 June 2021	*p*-Value
Total, *n* (%)	*n* = 151/151 (100%)	*n* = 126/151 (83.4%)	*n* = 25/151 (16.6%)	
Patients distribution by hospital, *n* (%)				0.4
Sepaco Hospital, Brazil	72/151 (48%)	58/126 (46%)	14/25 (56%)	
Béclère Hospital, France	79/151 (52%)	68/126 (54%)	11/25 (44%)	
Mean age, years mean (IQR)	31.0 (28.0, 35.0)	31.0 (28.0, 34.0)	33.0 (30.0, 36.0)	0.069
Ethnic group, *n* (%)				0.5
White	85/151 (56%)	70/126 (56%)	15/25 (60%)	
Black	27/151 (18%)	24/126 (19%)	3/25 (12%)	
Maghrebian	18/151 (12%)	13/126 (10%)	5/25 (20%)	
Hispanic	20/151 (13%)	18/126 (14%)	2/25 (8.0%)	
Asian	1/151 (0.7%)	1/126 (0.8%)	0/25 (0%)	
Pre-pregnancy BMI, kg/m^2^ mean, (IQR)	26.2 (23.3, 30.3)	26.0 (23.0, 29.9)	29.0 (26.0, 35.0)	0.011
Parity, *n* (%)				0.4
Nulliparous	73/151 (48.3%)	59/126 (46.9%)	14/25 (56.0%)	
Multiparous	78/151 (51.7%)	67/126 (53.1%)	11/25 (44.0%)	
Current smoking, *n* (%)	16/151 (11%)	13/126 (10%)	3/25 (12%)	0.7
Pre-existing medical conditions, *n* (%)	38/151 (25%)	29/126 (23%)	9/25 (36%)	0.2
Chronic hypertension	14/151 (9.3%)	13/126 (10%)	1/25 (4.0%)	0.5
Diabetes mellitus	1/151 (0.7%)	0/126 (0%)	1/25 (4.0%)	0.2
Pulmonary disease (including asthma)	11/151 (7.3%)	8/126 (6.3%)	3/25 (12%)	0.4
Other	12/151 (7.9%)	8/126 (6.3%)	4/25 (16%)	
Multiple pregnancies, *n* (%)	8/151 (5.3%)	7/126 (5.6%)	1/25 (4.0%)	>0.9
COVID-19 infection-related information				
Gestational age at the time of positive RT-PCR, weeks mean (IQR)	32 (26, 35)	31 (26, 35)	34 (32, 36)	0.12
Hospital admissions for COVID-19-related illness, *n* (%)	57/151 (38%)	43/126 (34%)	14/25 (56%)	0.039

BMI: body mass index; IQR: interquartile range.

**Table 2 jcm-11-02663-t002:** Disease severity among symptomatic pregnant women during the “wild-type period” and “variant period”.

Characteristic	Wild-Type Period	Variant Period	Univariate Analysis			Multivariate Analysis ^1^		
	126/151 (83.4%)	25/151 (16.6%)	OR	95%CI	*p*-Value	aOR ^1^	95%CI	*p*-Value
Primary Outcome								
Severe-to-critical infection according to WHO	22/126 (17%)	13/25 (52%)	5.12	2.07, 12.9	<0.001	7.07	2.50, 21.6	<0.001
Secondary Outcomes								
Admission to ICU	13/126 (10%)	8/25 (32%)	4.09	1.44, 11.3	0.007	3.75	1.22, 11.2	0.018
Oxygen support	21/126 (17%)	12/25 (48%)	4.62	1.84, 11.6	0.001	5.47	2.00, 15.7	0.001
Pneumonia	22/126 (17%)	11/25 (44%)	3.71	1.47, 9.30	0.005	3.94	1.43, 11.1	0.008
ARDS	7/126 (5.6%)	5/25 (20%)	4.25	1.16, 14.7	0.022	3.83	0.95, 14.6	0.050
Mechanical ventilation	6/126 (4.8%)	5/25 (20%)	5.00	1.33, 18.2	0.014	4.72	1.12, 19.6	0.030
ECMO	0/126 (0%)	0/25 (0%)	-	-	-	-	-	-
Maternal death	0/126 (0%)	1/25 (4.0%)	-	-	-	-	-	-
Disease severity					0.001			
Non-severe	103/126 (82%)	12/25 (48%)						
Severe	16/126 (13%)	8/25 (32%)						
Critical	7/126 (5.6%)	5/25 (20%)						
Hospital admission for COVID-19-related illness	43/126 (34%)	14/25 (56%)			0.039			

ARDS: acute respiratory distress syndrome; ECMO: extracorporeal membrane oxygen therapy; ICU: intensive care unit; WHO: World Health Organization; aOR: adjusted odds ratio; CI: confidence interval. ^1^ Odds ratio adjusted for age, obesity (BMI > 30), center, pre-existing condition status (yes or no), and geographic origin.

**Table 3 jcm-11-02663-t003:** Obstetrical outcomes among symptomatic pregnant women during the “wild-type period” and “variant period”.

Characteristic	Wild-Type Period	Variant Period	Univariate Analysis			Multivariate Analysis ^1^		
	1 February 2020–13 February 2021	14 February 2021–1 June 2021	OR	95%CI	*p*	aOR ^1^	95%CI	*p*
Composite adverse obstetric outcome	51/126 (40%)	15/25 (60%)	2.21	0.93, 5.44	0.077	1.86	0.73, 4.86	0.2
Secondary Obstetrical Outcomes								
Pre-eclampsia; eclampsia; HELLP	11/126 (8.7%)	2/25 (8.0%)	0.91	0.13, 3.68	0.91	0.11	0.01, 0.84	0.056
Gestational age at delivery, weeks mean (IQR)	39.0 (37.3, 40.0)	37.1 (34.1, 39.0)	-	-	0.045	-	-	-
<32 weeks	11/126 (8.7%)	1/25 (4.0%)	0.44	0.02, 2.40	0.44	0.35	0.02, 2.34	0.4
<37 weeks	25/126 (20%)	11/25 (44%)	3.17	1.27, 7.85	0.012	3.87	1.44, 10.6	0.007
Spontaneous delivery <37 weeks	7/126 (5.6%)	2/25 (8.0%)	1.48	0.21, 6.59	0.64	2.45	0.21, 23.7	0.4
Unscheduled cesarean	42/126 (33%)	10/25 (40%)	1.33	0.54, 3.19	0.52	1.10	0.42, 2.77	0.8
Suspected fetal distress	24/126 (19.0%)	6/25 (24.0%)	0.88	0.29, 2.38	0.80	0.45	0.12, 1.41	0.2
Postpartum hemorrhage	20/126 (16%)	0/25 (0%)	-	-	-	-	-	-
Stillbirth	3/126 (2.4%)	1/25 (4.0%)						
<24 weeks	1/126 (0.8%)	1/25 (4.0%)	5.21	0.20, 135	0.25	6.04	0.15, 690	0.3
>24 weeks	2/126 (1.6%)	0/25 (0%)	-	-	-	-	-	-
Deep venous thromboembolism/pulmonary embolism	0/126 (0%)	0/25 (0%)	-	-	-	-	-	-
Maternal death	0/126 (0%)	1/25 (4.0%)	-	-	-	-	-	-

HELLP: hemolysis, elevated liver enzymes, and low platelet count; IQR: interquartile range; aOR: adjusted odds ratio; CI: confidence interval. ^1^ Odds ratio adjusted for age, obesity (BMI > 30), center, pre-existing condition status (yes or no), and geographic origin.

**Table 4 jcm-11-02663-t004:** Neonatal outcomes among symptomatic pregnant women during the “wild-type period” and “variant period”.

Characteristic	Wild-Type Period	Variant Period	Univariate Analysis			Multivariate Analysis ^1^		
	1 February 2020–13 February 2021	14 February 2021–1 June 2021	OR	95%CI	*p*	aOR ^1^	95%CI	*p*
Composite adverse neonatal outcome	42/131 (32%)	10/24 (42%)	1.51	0.61, 3.67	0.36	1.42	0.55, 3.58	0.5
Secondary Neonatal Outcomes								
Weight, g (IQR)	3060 (2612, 3402)	2945 (2285, 3332)	-	-	0.3	-	-	0.6
Z-score	−0.52 (−1.23, 0.23)	−0.43 (−1.22, 0.25)	-	-	0.9	-	-	0.8
SGA (Z-score < −1.28)	28/128 (21.9%)	6/23 (26.1%)	0.89	0.20, 2.93	0.86	0.93	0.20, 3.29	>0.9
Admission to NICU	11/128 (8.6%)	8/23 (35%)	5.67	1.93, 16.4	0.001	4.94	1.37, 18.4	0.014
5′ Apgar score < 7	9/128 (7.0%)	0/23 (0%)	-	-	0.4	-	-	-
Arterial umbilical cord pH, mean (IQR)	7.24 (7.18, 7.28)	7.24 (7.21, 7.29)	-	-	0.6	-	-	0.7
Neonatal respiratory distress	10/128 (7.8%)	6/23 (26%)	4.16	1.28, 12.8	0.014	3.34	0.88, 12.2	0.068
Grade 3/4 intraventricular hemorrhage	3/128 (2.3%)	1/23 (4.3%)	1.89	0.09, 15.6	0.59	1.10	0.05, 10.3	>0.9
Neonatal death	4/128 (3.1%)	0 (0%)	-	-	-	-	-	-
Congenital malformation	2/128 (1.5%)	0/23 (0%)	-	-	-	-	-	-

aOR: adjusted odds ratio; CI: confidence interval; IQR: interquartile range; NICU: neonatal intensive care unit; SGA: small for gestational age; ^1^ adjusted for geographic origin, preterm birth, and center.

## Data Availability

The data presented in this study are available on request to the corresponding author.

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
