# Peer review of "Impact of SARS-CoV-2 Alpha and Gamma Variants among Symptomatic Pregnant Women: A Two-Center Retrospective Cohort Study between France and Brazil"

_jcm, 2022, doi:10.3390/jcm11092663_

Round 1
Reviewer 1 Report
This is a two-center retrospective cohort study which looked at obstetric outcomes when effected by wild variants vs alpha and gamma variants. Risk of admission to ICU was higher, needing oxygen support was higher, risk of pneumonia, ARDS, mechanical ventilation, maternal death and hospital admissions were higher in the period of feb 2020 till feb 2021 compared to feb 14 2021 till June 1st, 2021. As per the study, even the fetal outcomes like still births, fetal distress was also higher in 2021 compared to 2020.
It was concluded that, alpha and gamma variants were dangerous compared to wild variants. In France, the risk of developing severe infection was 3 times greater during alpha period than wild variant. While in Brazil, the risk was 12 times greater in gamma period than wild type variant.
We are not able to say for sure that these outcomes surely represent alpha and gamma variants as the patients were not individually tested. But there were similar studies which took in to consideration the dominant variant in that period. This is a good study but patient number and period of follow up was also only 3 months in variant group.
Author Response
Reviewer 1 – Review Report
- This is a two-center retrospective cohort study which looked at obstetric outcomes when effected by wild variants vs alpha and gamma variants. Risk of admission to ICU was higher, needing oxygen support was higher, risk of pneumonia, ARDS, mechanical ventilation, maternal death and hospital admissions were higher in the period of feb 2020 till feb 2021 compared to feb 14 2021 till June 1st, 2021. As per the study, even the fetal outcomes like still births, fetal distress was also higher in 2021 compared to 2020.
It was concluded that, alpha and gamma variants were dangerous compared to wild variants. In France, the risk of developing severe infection was 3 times greater during alpha period than wild variant. While in Brazil, the risk was 12 times greater in gamma period than wild type variant.
Thank you very much for this accurate summary of our manuscript.
- We are not able to say for sure that these outcomes surely represent alpha and gamma variants as the patients were not individually tested. But there were similar studies which took in to consideration the dominant variant in that period.
This is indeed one of the limitations of our study. Unfortunately, we could not obtain the identification of the strain for all the samples.
As you mentioned, several other articles focusing on Covid variants have worked on the basis of the periods of predominance of each variant because of their rapid spread, making each variant almost the only strain detected in each period. We have added this to the discussion (l.342): “This method is widely used by authors who have studied different strains of SARS-CoV-2 and more generally by those who study infectious disease.”
- This is a good study but patient number and period of follow up was also only 3 months in variant group.
The choice of the end of the data collection period is mainly linked to the emergence of the Delta variant in France in early June 2021. The explosive progression of this variant made it the major strain in France by the beginning of July 2021. (These French data were collected by Santé Publique France and are visible on SantéPubliqueFrance and GEODES). We decided to stop the data collection before the emergence of the Delta strain in order not to introduce another bias.
Reviewer 2 Report
The article I reviewed is very current and is very interesting for the medical staff. Knowing the association of Covid-19 with various diseases and side effects is very important.
Although the article submitted is well done, I consider that it needs to be improved by explaining the techniques used to detect viral subtypes. The laboratory method used is not very clear in the chapter.
Secondly, I think it is very important to analyze and corroborate the results obtained with the vaccine status. In the Discussions chapter it is true that an attempt is made to explain why these data were not analyzed, but I do not consider that the explanation is sufficient. The study group runs from February 14, 2021 to June 1, 2021. It is known that the vaccination rate was high during this period and the elderly population was vaccinated.
In conclusion, I consider that a reassessment of the results presented from the perspective of vaccine status is important.
Author Response
Reviewer 2 – Review Report
- The article I reviewed is very current and is very interesting for the medical staff. Knowing the association of Covid-19 with various diseases and side effects is very important. Although the article submitted is well done, I consider that it needs to be improved by explaining the techniques used to detect viral subtypes. The laboratory method used is not very clear in the chapter.
Thank you very much for your review. Perhaps we were unclear in our introduction. We did not directly study the incriminated variants for each of our patients since we did not have the information for most of the samples. We chose to study the periods during which each variant was the predominant strain. As explained in our introduction, in France, Santé Publique France sequenced bi-weekly sampled PCRs to give the distribution of each variant at the national level over time. In Brazil, different studies mentioned allowed us to know the distribution of each variant depending on time.
For more clarification:
- in the introduction we have added the following sentence (l.120-122): “The speed of installation of each variant and its rapid predominance combined with the fact that not all samples were sequenced pushed us not to study the variants directly but the corresponding periods.”
- in the discussion we have added the following sentence (342): “This method is widely used by authors who have studied different strains of SARS-CoV-2 and more generally by those who study infectious disease.”
- Secondly, I think it is very important to analyze and corroborate the results obtained with the vaccine status. In the Discussions chapter it is true that an attempt is made to explain why these data were not analyzed, but I do not consider that the explanation is sufficient. The study group runs from February 14, 2021 to June 1, 2021. It is known that the vaccination rate was high during this period and the elderly population was vaccinated.
In conclusion, I consider that a reassessment of the results presented from the perspective of vaccine status is important.
We fully agree that vaccination is now a fundamental parameter to be considered in studies investigating SARS-CoV-2. Vaccination did not really start for pregnant women until spring 2021 and even at the end of the study vaccination coverage was extremely low in both countries. We have added a more precise description of the vaccination data as of June 1st (source GEODES of Santé Publique France for France and WHO for Brazil) and added this paragraph to better detail the vaccination coverage (l. 345 to 351 and 2 more references): “In France, on June the 1st, less than 9% of women under 50 years old were fully vaccinated [27] and the recommendations for vaccination in pregnant women had only just been issued. In Brazil, on June the 1st of 2021, 10.6% of the population was fully vaccinated[28], and this concerned mainly pensioners and health professionals because of lack of accessibility to vaccination at this time.”
Reviewer 3 Report
Comments:
The topic of the present original study, evaluating the impact of SARS-CoV-2 Alpha and Gamma variants among symptomatic pregnant women: a two-center 3 cohort study between France and Brazil, appears deserving of attention.
Reported findings currently presented may pave the way for further investigations on a larger scale.
The study is well conducted and described and the manuscript is well organized and written.
Materials and Methods and Results are clearly presented. Introductions as well as Discussion sections are well structured.
Minor revision for English language required.
Concerns and suggestions:
Title:
Please define the study design in the title, for example “A Retrospective Study”
Abstract:
Please, add “in pregnant women” to the sentence “ The primary aim of this study was to compare the severity of 23 COVID-19 infection” (line 24).
Introduction:
Please, substitute “now” with “in the meanwhile” or similar (line 47);
Please, remove “us” (line 47);
Please, substitute “than for” with “compared with” (line 48);
Please, remove “more” repeated in lines 48 and 49;
I would suggest to start a new paragraph with the period beginning in line 68.
Results:
Please, add “among the 238 SARS-Cov-2 PCR positive pregnant women” to “we identified 151 patients” (line 194);
Figure 1: modify the first box “SARS-Cov-2 PCR positive pregnant women between ….”;
Lines 203-203: I would suggest to modify the period “For each period, recruitment 203 was approximately 50% of patients” in “For each period, based on eligibility criteria, approximately the 50% of the patients resulted eligible and were thus considered for the present study”.
Table 1: Should heading “COVID-19 infection related information” be moved forward?
Author Response
Reviewer 3 – Review report
Comments:
- The topic of the present original study, evaluating the impact of SARS-CoV-2 Alpha and Gamma variants among symptomatic pregnant women: a two-center 3 cohort study between France and Brazil, appears deserving of attention.
Reported findings currently presented may pave the way for further investigations on a larger scale.
The study is well conducted and described and the manuscript is well organized and written.
Materials and Methods and Results are clearly presented. Introductions as well as Discussion sections are well structured.
Minor revision for English language required.
Thank you very much for your review and your accurate comments.
Concerns and suggestions:
Title:
- Please define the study design in the title, for example “A Retrospective Study”
We have taken your comment into account and have modified the title as follows: “Impact of SARS-CoV-2 Alpha and Gamma variants among symptomatic pregnant women: a two-center retrospective cohort study between France and Brazil”
Abstract:
- Please, add “in pregnant women” to the sentence “ The primary aim of this study was to compare the severity of 23 COVID-19 infection” (line 24).
Done.
Introduction:
- Please, substitute “now” with “in the meanwhile” or similar (line 47);
Done.
- Please, remove “us” (line 47);
Done.
- Please, substitute “than for” with “compared with” (line 48);
Done.
- Please, remove “more” repeated in lines 48 and 49;
Done.
- I would suggest to start a new paragraph with the period beginning in line 68.
Done.
Results:
- Please, add “among the 238 SARS-Cov-2 PCR positive pregnant women” to “we identified 151 patients” (line 194);
Done.
- Figure 1: modify the first box “SARS-Cov-2 PCR positive pregnant women between ….”;
Done.
- Lines 203-203: I would suggest to modify the period “For each period, recruitment 203 was approximately 50% of patients” in “For each period, based on eligibility criteria, approximately the 50% of the patients resulted eligible and were thus considered for the present study”.
Our sentence was unclear. What we wanted to say is that there were approximatively the same number of patients included in each of the study centers for the two periods. We have made the following change in consequence: (l.215) “For each period, approximately the same number of patients were included at each of the two study centers.”
- Table 1: Should heading “COVID-19 infection related information” be moved forward?
We have modified the heading accordingly.
Round 2
Reviewer 2 Report
The answers sent by the authors I think answer the problems raised by the reviewers. The additions brought in the article bring new clarifications and are part of the idea of a better understanding of this article.